# Petroleum exploration increases methane emissions from northern peatlands

Maria Strack[1], Shari Hayne[2], Julie Lovitt [3], Gregory J. McDermid [3], Mir Mustafizur Rahman [3], Saraswati Saraswati[1] & Bin Xu[4]

Peatlands are globally significant sources of atmospheric methane ($CH_4$). In the northern hemisphere, extensive geologic exploration activities have occurred to map petroleum deposits. In peatlands, these activities result in soil compaction and wetter conditions, changes that are likely to enhance $CH_4$ emissions. To date, this effect has not been quantified. Here we map petroleum exploration disturbances on peatlands in Alberta, Canada, where peatlands and oil deposits are widespread. We then estimate induced $CH_4$ emissions. By our calculations, at least 1900 km$^2$ of peatland have been affected, increasing $CH_4$ emissions by 4.4–5.1 kt $CH_4$ yr$^{-1}$ above undisturbed conditions. Not currently estimated in Canada's national reporting of greenhouse gas (GHG) emissions, inclusion would increase current emissions from land use, land use change and forestry by 7–8%. However, uncertainty remains large. Research further investigating effects of petroleum exploration on peatland GHG fluxes will allow appropriate consideration of these emissions in future peatland management.

[1] Department of Geography and Environmental Management, University of Waterloo, Waterloo, ON N2L 3G1, Canada. [2] Science and Technology Branch, Environment and Climate Change Canada, Gatineau, QC K1A 0H3, Canada. [3] Department of Geography, University of Calgary, Calgary, AB T2N 1N4, Canada. [4] Boreal Research Institute, Northern Alberta Institute of Technology, Peace River, AB T8S 1R2, Canada. Correspondence and requests for materials should be addressed to M.S. (email: mstrack@uwaterloo.ca)

Globally, northern peatlands cover ~4 million km$^2$ and store 500 ± 100 Gt of soil carbon[1,2]. Saturated soils enable this accumulation of organic matter, but also methane (CH$_4$) production such that northern peatlands contribute ~36 Tg CH$_4$-C yr$^{-1}$ to global CH$_4$ emissions[3]. In the Canadian province of Alberta, oil sands (bitumen) deposits cover 142,200 km$^2$, and represent the world's third largest oil reserve[4]. Alberta also has important conventional oil and natural gas and shale gas deposits[4]. These landscapes also contain areas of dense peatland cover[5].

Seismic exploration is used for mapping geologic formations to determine economically viable deposits of natural resources, including oil and gas. In Alberta, seismic exploration has been ongoing since 1929[6], and involves the placement of geophones on the surface to record energy reflected from underlying rocks layers[7]. Placement of these geophones requires the clearing of long linear trails across the landscape. Until the early 1990s, these lines were often cleared by bulldozers, creating 5–10 m wide linear features (hereafter seismic lines[8]). Since then, these legacy lines have been replaced by low-impact seismic (LIS) lines in Alberta and elsewhere in Canada (e.g., British Columbia, Yukon, Northwest Territories), with a reduced width (1.5–5 m) and slightly meandering path designed to reduce ecological impact[9]. However, LIS lines are generally placed in a high-density grid, often only tens of meters apart[8,10]. Although construction of lines prior to 1960 often resulted in substantial soil disturbance, improved management practices, including LIS, has greatly reduced the damage to soil and ground layer vegetation[10]. Despite this, even decades after seismic lines are created, many remain visible on the landscape, particularly when crossing wetland ecosystems[8,11].

While studies on the impact of seismic exploration on wildlife have been extensive[12–15], less is known about the effect on soil conditions[10]. Wetlands cover more than 50% of Alberta's oil sands in many areas, with peatlands accounting for over 90% of these wetlands[10,16]. In Canada, a peatland is defined as a wetland ecosystem in which at least 40 cm of organic soil has accumulated[17]. A best practice that has been developed to minimize soil disturbance in the peatland rich landscape is to conduct seismic exploration on frozen ground[10]. However, vegetation clearing and movement of heavy equipment on peatland still result in the removal of trees and disturbance to soil and ground vegetation[18,19]. There is evidence of soil compaction when lines have been used repeatedly[20] and lowering of the surface elevation and flattening of microtopography even on LIS lines in peatlands[21]. This results in persistent changes in vegetation community[20,22,23] and shallower water table (WT) position[20,21,24]. Soils on peatland seismic lines also become warmer, with thicker active layers in permafrost zones[23,24], likely due to increased incident solar radiation once the canopy is removed, and more importantly higher thermal conductivity of saturated soils[25]. As northern peatland CH$_4$ emissions are driven by WT, temperature and vegetation community[26], altered conditions present on seismic lines are likely to enhance CH$_4$ emissions compared to undisturbed peatland area (Fig. 1). The objective of this study is to estimate the impact of seismic lines on peatland CH$_4$ emissions across the province of Alberta. We then use these estimates to speculate on the impact of petroleum exploration on land use CH$_4$ emissions more broadly, including Canada's national GHG accounting.

Based on our calculations, we show that seismic lines disturb at least 1900 km$^2$ of peatlands in Alberta, with LIS lines likely underreported in this estimate. Shallower WT on these lines enhances peatland CH$_4$ emissions by up to 5.1 kt CH$_4$ above undisturbed conditions. However, uncertainty in these emissions remains large due to the dearth of measurements of environmental conditions and GHG fluxes from peatlands affected by petroleum exploration, limiting our process-based understanding of ecosystem response. Given the extent of peatland areas affected, research programs further investigating the potential effects of petroleum exploration on peatland GHG fluxes are required to allow for appropriate consideration of these emissions in the planning of future exploratory activities and peatland restoration programs.

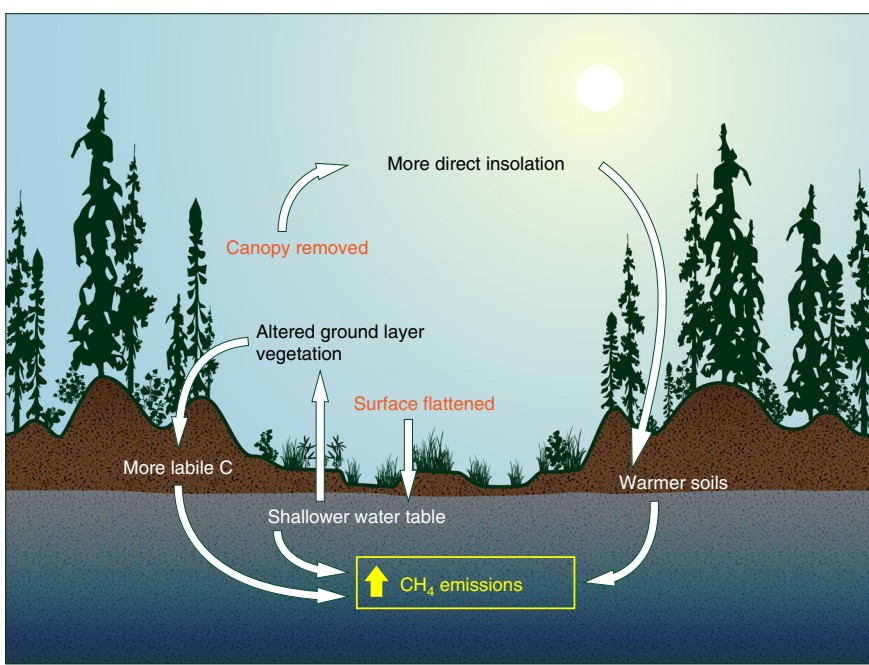

**Fig. 1** Conceptual model of methane flux on peatland seismic lines. Included here are ecohydrological changes that occur on peatland seismic lines that have the potential to increase methane emissions. Direct effects caused by exploration activities are shown in orange, with indirect effects arising from these in black/white

## Results

**Area of peatland impacted by seismic lines.** In order to estimate the potential impact of seismic exploration on $CH_4$ emissions in Alberta, we used data on human disturbances from the Alberta Biodiversity Monitoring Institute's (ABMI) Human Footprint Inventory[27] and information on wetlands from Alberta Environment and Parks' Merged Wetland Inventory[28] (Fig. 2; see details in Methods). Here, we include bogs and fens in the Alberta wetland inventory, as these are defined as peatlands in the Canadian Wetland Classification System[17]. We also estimate the impact on swamps, defined here as wetlands with dominant tall woody vegetation, normally with greater than 30% canopy cover and sometimes accumulating enough organic material to be considered a peatland[17]. Soil properties for swamps in western

Canada[29] indicate that most have a thick enough organic layer to meet the peatland definition, so we include them here as peatlands; some may be mineral soil wetlands leading to some overestimation of total seismic line length in peatlands. Based on these data sets, we estimate that at least 345,000 km of seismic lines and trails (measured here as length; see Methods for complete definitions) cross peatlands in Alberta, of which about 10% are LIS (Table 1).

Considering average widths of each disturbance type (see Methods), the area of peatland disturbed by seismic lines and trails in Alberta is over $1900 \, km^2$ (see details in Methods). We expect these figures to be lower than the true amount of disturbance. While the ABMI Human Footprint Inventory data is the best source of publically available information, its

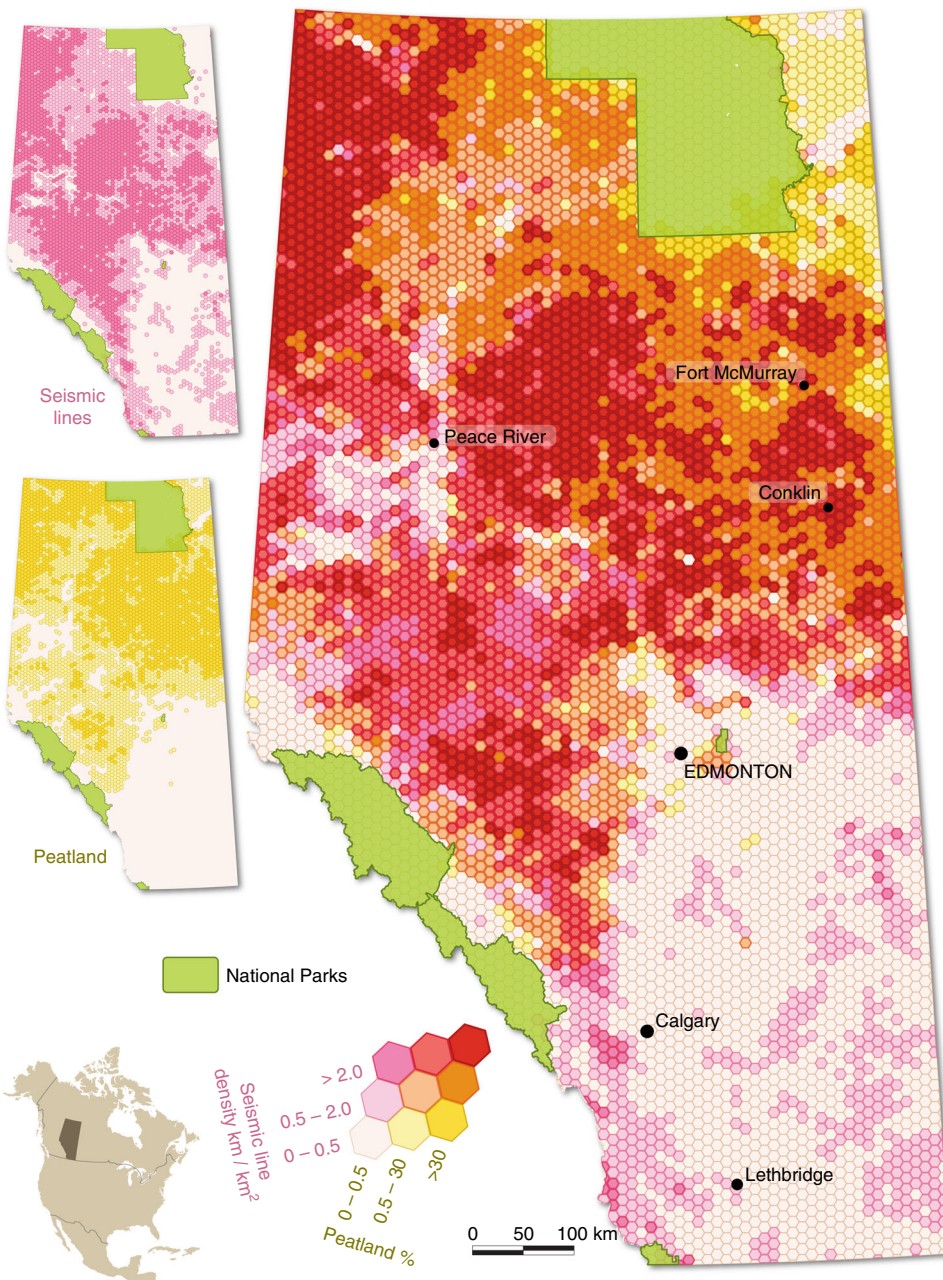

**Fig. 2** Peatland and seismic line density. This series of maps illustrates the spatial overlap of peatlands and seismic lines in Alberta, Canada. Overview #1 (Seismic lines) displays the spatial distribution of seismic lines across the province, measured in terms of density. Overview #2 (Peatland) shows the distribution of peatlands across the province, measured in terms of percent cover. In the main map, hot colors indicate where high densities of seismic lines and high proportions of peatlands coincide

**Table 1 Summary of peatland and seismic line area and impact on methane emission**

| Peatland type | Bog | Fen | Swamp | Total[a] |
|---|---|---|---|---|
| Total area in Alberta (km$^2$) | 30,050 | 58,580 | 46,160 | **134,790** |
| Total length of seismic lines[b] (km) | 88,580 | 144,580 | 112,480 | **345,640** |
| Total area[c] covered by seismic lines (km$^2$) | 490 | 790 | 630 | **1910** |
| Average CH$_4$ undisturbed (g CH$_4$ m$^{-2}$)[d] | 1.1 | 3.4 | 0.40 | |
| CH$_4$ flux on seismic line (g CH$_4$ m$^{-2}$)[d] | 2.5–2.8 | 7.8–8.5 | 0.80–0.85 | |
| Enhanced CH$_4$ flux due to seismic lines (kt CH$_4$) | 0.69–0.80 | 3.44–4.01 | 0.25–0.29 | **4.43–5.14** |

[a]Values may not add to total due to rounding
[b]Total of conventional, low-impact seismic lines and trails. See Methods and Supplementary Tables 2 and 3 for length and area of each line type
[c]Estimated based on average widths of each line type as outlined in Methods
[d]Estimated from mean daily fluxes during summer extended over the 123-day growing season. Winter fluxes assumed negligible. See Methods for more information

mapping protocol relies extensively on satellite imagery that likely misses many small disturbances such as LIS[28]. To better understand the potential extent of this underestimation, we compared the publically available ABMI Human Footprint Inventory to an enhanced linear-feature dataset provided by ABMI[30], which was generated using high-resolution imagery (see Methods). We found that 70% of the LIS lines present in a 10,000 km$^2$ sample area in central Alberta are not captured in the publically available dataset, suggesting that we might be substantially underestimating this type of disturbance in our analysis (Fig. 3; values for all seismic line types presented in Supplementary Table 1). Overall, only 53% of the total length of seismic lines and trails crossing peatlands in this test area were included in the ABMI Human Footprint Inventory.

**Impact of seismic lines on peatland CH$_4$ emissions.** Regression equations between log-transformed CH$_4$ flux and WT position were statistically significant ($p < 0.05$) for each peatland type (Fig. 4). Based on our compiled database of peatland CH$_4$ flux and WT from western Canada (see Supplementary Data 1), we determined a mean WT position of $-18$, $-8$, and $-22$ cm for undisturbed swamps, fens, and bogs, respectively (where negative values indicate distance below the ground surface) leading to estimated mean emissions of 1.4, 7.1, and 2.5 g CH$_4$ m$^{-2}$ yr$^{-1}$. These values are on the low end of ranges of CH$_4$ fluxes reported in recent compilations of northern peatland data[3,26]—not surprising given the dry continental climate in western Canada. Using these mean values, we estimated province-wide CH$_4$ emissions from peatlands at 251 kt (95% confidence interval: 0–3200 kt) without any disturbance.

We estimated the potential increase in CH$_4$ emission related to seismic lines based on changes in WT position: a well-documented relationship in peatlands[3,26,31]. Starting with the mean WT estimated for each peatland type (Fig. 4), we then moved the mean WT position closer to the surface on the footprint of the seismic lines according to the change in WT previously reported in literature (13.9–15.4 cm[20,21]) and re-calculated CH$_4$ flux (see Methods for more details). Based on this, we estimate that seismic lines increase peatland CH$_4$ emissions in Alberta by 4.4–5.1 kt CH$_4$ yr$^{-1}$ (Table 1).

**Discussion**
Currently, there are very few studies on the effect of seismic lines on peatland carbon and GHG exchange[10] or how this will contribute to anthropogenic radiative forcing[32]. We have chosen to use WT position to estimate the potential impact as the shift in WT on peatland seismic lines has been reported for both bogs[21] and fens[20]. The former study mapped average WT changes over a 61 ha area, providing more confidence that these represent more than local conditions. However, seismic lines likely also alter

thermal and ecological conditions (Fig. 1), which are known controls on peatland CH$_4$ emissions. Strack et al.[20] found greater C uptake and an order-of-magnitude increase in CH$_4$ emissions on a 6 m wide seismic line that had been converted to a winter road. This increase in CH$_4$ emission was attributable to higher soil temperature and a shift toward a graminoid-dominated plant community that likely provided a labile carbon source and acted as a conduit for CH$_4$ transport (Fig. 1). Other studies have shown width, age, and orientation to influence thermal and hydrological conditions on linear disturbances[24]. It is clear that more research is needed to better quantify actual changes in CH$_4$ emissions under the variety of disturbance conditions that occur on peatland seismic lines and enable future work to estimate emissions using process-based models. As we considered only hydrological changes, and not shifts in temperature regimes or vegetation communities, our calculated value likely underestimates enhanced CH$_4$ emissions, but again, more field data is needed to confirm this. In addition, construction of seismic lines may have an impact on the adjacent peatland[33] (i.e., edge effects) that could further increase GHG impact of linear disturbances. This, combined with the known underestimation of LIS line area in the human footprint database (Fig. 3), further point to the likely underestimation of impact from the present analysis.

We also included swamps in our estimates of seismic line impacts to peatlands, although some are likely mineral soil wetlands. Assuming both mineral soil and peatland swamps respond similarly to linear disturbances, our estimate of increased CH$_4$ emissions from swamps would not overestimate the impact in emissions, but may misclassify some wetland impact as specific to peatlands. Since there is no available data on the hydrologic impact of seismic lines in swamps, the potential uncertainty of including mineral soil swamps in our estimates is unclear. However, studies of tree regrowth on seismic lines also indicate poor recovery in swamp ecosystems[11], suggesting that including them in our analysis is warranted.

We have estimated land use CH$_4$ emissions from peatland seismic lines over the province of Alberta, Canada, a location where petroleum exploration has resulted in at least 345,000 km of seismic-line disturbance on boreal peatlands. Given some basic assumptions concerning the average widths lines, the disturbed area in the province is over 1900 km$^2$: more than five times that of the heavily regulated peat-extraction industry across all of Canada[34]. Similar seismic exploration has also occurred across large regions of boreal and subarctic North America. For example, the average density of linear disturbance is 0.46 km/km$^2$ in the boreal plains ecozone of the province of Saskatchewan[35]. Assuming 30% wetland cover across Saskatchewan's boreal forest[36], this represents potentially an additional 15,000 km of peatland linear disturbance[36]; again, this likely underestimates the presence of LIS lines[37,38]. Seismic lines are also abundant in Canada's Northwest Territories[24], while winter roads are present

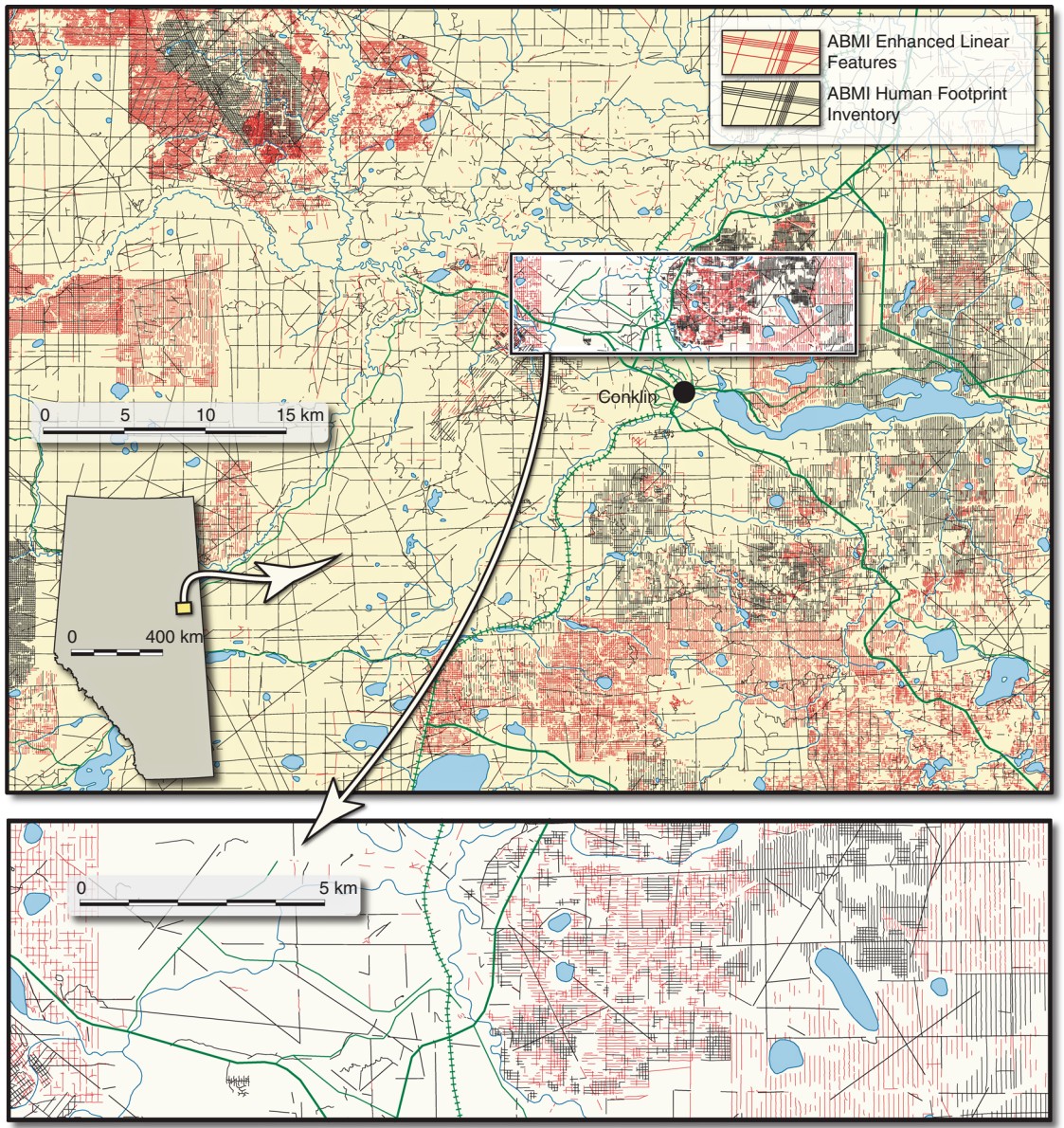

**Fig. 3** Missing seismic lines in the publicly available database. This map shows a comparison of seismic lines and trails captured in the publicly available Human Footprint Inventory[26] to those in an Enhanced Linear Feature database provided by the Alberta Biodiversity Monitoring Institute[30]. Our estimates of induced methane emissions are based on the Human Footprint Inventory, shown in black. Any seismic lines present in the Enhanced Linear Feature database that did not occur in the Human Footprint Inventory are shown in red. Therefore all red seismic lines in the figure would be missed in the province-wide analysis and thus represent the potential for underestimation of the impact. While low-impact seismic (LIS) lines are not specifically symbolized on the map, they are relatively easy to distinguish from conventional networks on account of their very high density: often spaced less than 100 m apart. Across this 10,000 km² scene, 70% of the LIS lines are missing from the publicly available dataset. For the sake of comparison, the hexagons in Fig. 2 (which includes the town of Conklin) are 100 km² each

across Canada's boreal zone such that linear disturbances represent the largest human impact in the region[38]. Seismic exploration is also widespread in Siberia, a region with abundant peatland cover[39]. Therefore, oil and gas exploration, along with other linear disturbance such as winter roads, may have extensive unreported impacts on peatland $CH_4$ emissions worldwide.

Given the lack of knowledge regarding the effect of seismic line construction on soil-carbon exchange, there is no specific guidance from the Intergovernmental Panel on Climate Change for estimating GHG emissions caused by this type of land use activity[40]. As many of the trees on peatlands are non-merchantable, vegetation was cleared, windrowed and slashed with historic clearing practices, and since the mid-2000s was coarse-

mulched and left to decay. Therefore, only emissions from harvesting wood in upland soils during seismic line clearing and the subsequent use of harvested wood products are included in national GHG inventory estimates[40]. While our estimate of peatland seismic line impact on $CH_4$ emissions, 4.4–5.1 kt yr$^{-1}$ is small compared to the ~4000 kt yr$^{-1}$ of $CH_4$ emitted from all Canadian peatlands[41], the present study represents emissions only from the province of Alberta, and this total is not negligible in terms of national $CH_4$ emissions arising from land use. Current anthropogenic $CH_4$ estimates for the land use, land use change and forestry sector of Canada's National GHG Inventory in 2016 are 63 kt $CH_4$[34]. This estimate represents $CH_4$ emissions from burning of agricultural grassland, burning of forest residues after

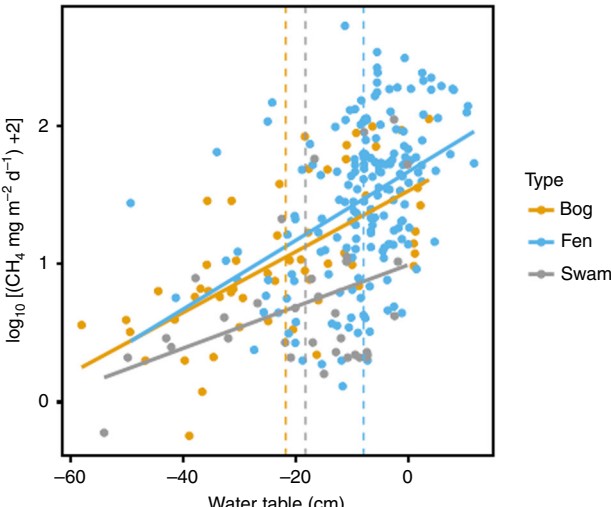

**Fig. 4** Methane flux versus water table position. These regressions were used to estimate the impact of seismic lines on methane ($CH_4$) flux. All regression lines are significant: bog—$\log CH_4 = 0.022$(water table) $+ 1.53$, $R^2 = 0.42$, $F_{1,55} = 39.63$, $p < 0.001$; fen—$\log CH_4 = 0.025$(water table) $+ 1.67$, $R^2 = 0.17$, $F_{1,174} = 35.81$, $p < 0.001$; swamp—$\log CH_4 = 0.015$(water table) $+ 0.99$, $R^2 = 0.16$, $F_{1,32} = 6.08$, $p = 0.019$. Mean water table for each peatland type is shown with a dashed vertical line and was used to calculate mean $CH_4$ emissions for undisturbed areas. On seismic lines, water table was moved towards the surface by 13.2 and 15.4 cm to calculate $CH_4$ emissions on disturbed areas (see Methods)

harvesting or deforestation and drainage and rewetting due to peat extraction. Including the $CH_4$ emissions from seismic lines on peatlands estimated in the present study would increase $CH_4$ emissions due to land use, land use change and forestry by 7–8%. However, given our underestimation of both the disturbed area and $CH_4$ flux on the lines, the impact is likely much higher. Moreover, given that seismic lines in peatlands persist for decades[11], these elevated emissions likely also remain over the same time scales.

The Paris Agreement highlighted the importance of accelerating the reduction of global anthropogenic GHG emissions and recognized the importance of conserving and enhancing sinks and reservoirs of GHGs as appropriate[42]. Therefore, there is an urgent need to better quantify the effect of seismic exploration on peatland carbon and GHG exchange. We recommend employing a multi-scale approach aimed at determining local factors driving changes in carbon uptake and $CH_4$ emissions, combined with improved mapping of shifts in ecohydrological conditions over regional scales. Trends in the area of peatland disturbed by seismic lines over time are also required. With LIS comprising much of the ongoing disturbance, accurate estimation will likely require increased access to industrial disturbance records: an issue mediated by provincial governments in Canada. Moreover, improved mapping of swamps and studies to characterize carbon exchange and impact of disturbance on this wetland type is needed to reduce uncertainty related to peatland seismic line impacts. Recent compilations of GHG flux data from Canadian peatlands also note the need for more $CH_4$ flux measurements from under-represented areas, including the Boreal Plains in which most Alberta peatlands are located[41]. Efforts on these fronts will enable the development of emission factors and activity data to improve the accuracy and completeness of national reporting of anthropogenic GHG emission estimates related to land use, as well as provide the process-based understanding needed to model and mitigate these emissions. Alberta

recently adopted restoration guidelines for legacy seismic lines, with the goal to mitigate wildlife impacts[10]. In wetlands, these activities focus largely on creating surface mounds on which trees can be planted in an attempt to speed up forest establishment[10]. Our findings indicate that without restoration, seismic lines crossing peatlands in Alberta contribute at least an additional 4.4–5.1 kt $CH_4$ to the atmosphere annually. Whether these restoration efforts will also meet the goals of reducing peatland GHG emissions remains unclear, highlighting the need for improved understanding of the effects of seismic lines on peatland function prior to the extensive application of restoration techniques.

## Methods
**Provincial peatland coverage**. The wetland inventory data for the Province of Alberta was obtained from the Alberta Merged Wetland Inventory (AMWI), prepared by the Environment and Parks Department, Government of Alberta, Canada[28]. The AMWI is provided as a vector polygon data set with five basic wetland classes: bog, fen, swamp, marsh, and open water. For this inventory, bogs, fens, and some classes of swamps occur in peatlands, while marshes, shallow/open water, and other classes of swamps occur in mineral soil wetlands[17]. We considered all area covered by bog, fen, and swamp as peatland, although this likely leads to overestimation of total peatland area as not all swamps will have the required 40 cm of organic matter accumulation under the Canadian peatland definition[17].

The AMWI is comprised of 33 separate inventory components generated by various organizations using different data sources, standards, timelines, and methods[28]. These data are later combined together and reclassified to above-mentioned classes to produce a single database for the entire province of Alberta. Thus, the product's inherent, internal inconsistencies and data gaps might limit its practical and reliable use for rigorous wetland monitoring.

**Provincial seismic line coverage**. The seismic line data (GIS Layer) was obtained from Alberta Biodiversity Monitoring Institute's (ABMI) Wall-to-Wall Human Footprint Inventory database[27] that depicts anthropogenic disturbances across the province in 2014. ABMI is entitled to collect, update (every 2 years) and distribute data on human footprint across the province of Alberta, Canada under the Alberta Human Footprint Monitoring Program (AHFMP).

The ABMI used Alberta Base Features as the basis for the seismic line product. Base Features is a GIS-ready dataset, containing 20+ baseline land cover features that have been complied internally within the Provincial Government of Alberta since 1996 for public use[27]. The seismic line layer within this database was updated by ABMI by manually interpreting SPOT6 satellite imagery. Three types of lines are included in the dataset; Trails, Legacy, and LIS lines. Trails can be industrial or recreational. Some trails are generated from abandoned linear features such as roads or old fire guards as well as detours from seismic lines. Since trails may arise from seismic lines and are difficult to discern from legacy seismic lines, we included them in our total seismic line estimate. Legacy lines are seismic lines that were constructed prior to the use of LIS construction methods. The change to LIS occurred between the late 1980s and early 1990s. The legacy lines were constructed using older technology that required the lines to be between 5 and 10 m in width[10] to allow equipment such as vibrator trucks or drilling equipment to operate on the lines. Seismic programs using the LIS construction methods have significantly narrower disturbances and often use avoidance construction methods and/or hand-cut lines to reduce not only the width of the line but also making them more sinuous to reduce line of site and better emulate natural patterns. This is possible in part due to advances in surveying with GPS, and also from the use of smaller drilling and vibrator equipment. Based on assumptions included in the seismic line dataset, we use average line widths of 6, 3, and 4 m for legacy lines, LIS, and trails, respectively.

**Provincial seismic line coverage within peatlands**. The provincial wetland data (polygon features) and the provincial seismic lines (line features) were intersected to identify seismic lines that fall within wetland. Linear length of different types of seismic line falling in different classes of peatland was then calculated from this intersected layer (Supplementary Table 2). In the next step, the lines were converted to polygons by buffering around the line to obtain corresponding line widths described above (see "Provincial seismic line coverage"). This polygon layer was used to calculate the area of different types of seismic lines within different types of wetlands (Supplementary Table 3) and create Fig. 2.

**Uncertainty in seismic line area in peatlands**. Seismic lines are difficult to map using satellite imagery due to their narrow widths, but manual delineation using relatively high resolution SPOT6 imagery provides the currently best existing database. Despite its strengths, there is a number of uncertainties associated with the ABMI provincial seismic line inventory. Firstly, this dataset represents mainly a

visible footprint from the sky. SPOT6 images are unable to locate narrow features (<~1.5 m). Therefore, many LIS lines that are present on the ground are not included in this dataset. Secondly, the distinction between trails, LIS and legacy seismic line features is based on human interpretation. Therefore, misclassification might have occurred in some cases. Thirdly, a general width of these lines is estimated. However, the actual on-ground widths can vary significantly, adding uncertainty to estimates of area disturbed. Fourthly, in the case of LIS lines, some areas remain to be updated by ABMI using SPOT6 imagery[26]. Finally, the seismic line and trail data might contain abandoned roads and fire guards that will have wider disturbances.

All these factors contribute to underestimation of seismic line coverage at the provincial level. To get a notion of how much it is underestimated, we compared our estimates of peatland seismic line length from the ABMI Human Footprint database to a higher resolution data set under development at ABMI (ABMI Enhanced Linear Features)[30], using mapped seismic line data with 1 m spatial resolution air photos in a ~10,000 km$^2$ site located near the town of Conklin, Alberta, Canada (55.6314° N, 111.0839° W). Differences in seismic line length between the datasets were computed (Supplementary Table 1).

**Methane emissions from Alberta peatlands.** We compiled literature values for methane ($CH_4$) flux from study sites in boreal continental western Canada (Manitoba, Saskatchewan, Alberta, Northwest Territories[16]) for bogs and fens. Although our case study focused on Alberta, we included data from a broader geographic region due to the scarcity of peatland $CH_4$ flux measurements in Alberta. As we could find very few records of $CH_4$ flux for swamps in western Canada, all Canadian sites for swamps were used. Given the low number of data points for swamps in general, it is unclear how the inclusion of eastern Canadian sites in the dataset impacts our estimates of swamp $CH_4$ flux. Data were compiled based on recent meta-analyses of global peatland $CH_4$ flux[3,43], a literature search of Web of Science using the terms, peatland methane Canada, and, swamp methane Canada, and existing flux measurements within our own research group, even if unpublished. In all cases, fluxes needed to be measured at least monthly in the summer period (May–August) and WT data reported in order to be included in the database. If data was only presented in a figure, numerical values were extracted from the image using WebPlotDigitizer[44]. When data from multiple sample plots were present in the data set, we took each plot as an individual data point if WT was also measured at each plot. As our goal was to develop a $CH_4$ flux-WT regression equation, we chose to maintain this plot-based data as it represents the scale at which WT and $CH_4$ vary spatially in peatlands. We used the mean $CH_4$ and WT during the measurement period, which included only the warm season (usually May–August/September). All values[20,21,45–59] included are given in Supplementary Data 1.

For each peatland class (i.e., bog, fen, swamp) we created a separate regression equation with $\log_{10}(CH_4 + 2)$ flux of the mean daily values as the dependent variable and mean WT position of the same measurement period as the independent variable in R[60]. The log-transformed data was used to improve normality and homogeneity of the residuals of the regression. Based on our compiled data set, we estimated the mean summer WT position for each peatland class. Using this WT position and the regression equation, we estimated the mean $CH_4$ flux for each class. We converted this value to an estimate of annual $CH_4$ emissions assuming a 123-day emission period (May–August). This is similar to growing season length estimated for sites in boreal Alberta when freezing temperatures are used as a threshold for start and end of the growing period[52]. While emissions may also continue during the dormant period, there are no reported wintertime measurements for western Canada and so we assume that emissions are negligible, acknowledging that this will underestimate total emissions. As our goal was to estimate the impact of seismic lines and not accurately determine provincial emissions, this underestimation has little impact on the conclusions.

Next, we moved the mean WT position closer to the surface of the peatland by 13.9 cm[20] and 15.4 cm[21] based on the measured hydrologic impact of seismic lines on peatlands in Alberta. We then recalculated the new $CH_4$ emission from the area of seismic lines occurring on each peatland class (as reported in Supplementary Table 3) and estimated a new annual peatland $CH_4$ emission for the province. The difference between this estimate and the original estimate based only on mean WT is reported as the potential effect of seismic lines on peatland $CH_4$ flux.

## Data availability

Data for wetland area and seismic lines coverage are publicly available. The Alberta Merged wetland inventory can be obtained at https://geodiscover.alberta.ca/geoportal/catalog/search/resource/details.page?uuid=%7BA73F5AE1-4677-4731-B3F6-700743A96C97%7D. The Alberta Biodiversity Monitoring Institute's Wall-to-Wall Human Footprint Inventory is available at https://abmi.ca/home/data-analytics/da-top/da-product-overview/GIS-Land-Surface/HF-inventory.html. Methane flux and water table data used in the analysis is included as Supplementary Data 1.

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

## Acknowledgements

The authors thank Shell Canada Ltd., Canadian Natural Resources Ltd., Emissions Reduction Alberta, the Natural Sciences and Engineering Research Council of Canada, and the Canada Research Chairs Program for funding. Robin Poitras and Jennifer Hird assisted in figure preparation. The authors thank the Alberta Biodiversity Monitoring Institute for providing samples of an Enhanced Linear Features inventory, which allowed them to confirm the suspected underestimation of LIS lines in publicly available data.

## Author contributions

M.S., J.L., G.J.M., M.M.R., S.S., and B.X. conceived of the study. S.H. calculated seismic line contribution relative to Canadian land use emissions. M.M.R. and G.J.M. determined peatland seismic line areas and calculated the underestimation of seismic lines in public databases. M.S. compiled the methane emission database and determined province-wide emission estimates. All authors contributed to manuscript preparation.

## Additional information

**Competing interests:** The authors declare no competing interests.

