## [Peer Review File · Nature Communications]

Reviewers' comments:

Reviewer #1 (Remarks to the Author):

General comments for Authors

The study addresses the effects of oil and gas exploratory activities (through construction of seismic lines) on methane emissions in northern peatlands. The place of study is in the province of Alberta, Canada, where both peatlands and seismic lines are highly abundant. As such, Alberta is a good choice for this study, but I also think that the fact that Alberta is so rich in oil and gas reserves (third highest in the world) and so densely disturbed by seismic lines, also makes it not necessarily an accurate representative of what is happening in the rest of Canada (or the rest of northern hemisphere for that matter). This should be recognized, if the study is to be used in providing information to be used at Canadian and/or global scale (e.g. for reporting GHG emissions). Still, I think that the study addresses a very important matter, which has not been adequately addressed before. I agree that there is not a lot of research (or the results of it are not easy to find/access), which addresses the effects of seismic lines on C, and GHG emissions in particular. Considering the abundance of northern peatlands, their importance as C sinks/sources, and the intensity of industrial disturbances on these peatlands, (including seismic lines, which are among the most prominent industrial linear disturbances), I think that the subject of this study is highly pertinent in the accurate assessment of GHG emissions and their effect on global climate change.

The methodology of the study has some shortcomings, which were nevertheless recognized and discussed (e.g. underestimation of the amount of LIS lines). Yet, generally I think that the resources available were used to the best of their potential. However, the study is based on previously known relationship of water table to CH₄ emissions only. While this relationship may be accurate enough, other important factors that have not been considered (soil temperature, plant community etc.) could have a different (even opposite) effects, which may potentially mitigate the effects of increased water table under seismic lines. Actual field measurements on and off seismic lines (including consideration of other than only water table factors) seem like the next logical step to corroborate the results of this study.

Statistical methods seem appropriate, though more precise information on which R version, package(s) and functions were used would be appreciated. Explanation of why the CH₄ flux and WT data was log transformed (presumably because of normality and homogeneity of variance assumptions violations?) would be good too. Supplementary information provided valuable additional information.

Overall, the manuscript is well-written: concise but comprehensive. However, as I was reading the manuscript, in quite a few instances I had a question/concern in the Results section, which was then answered/addressed later on, under the Methods section. Why is the Methods section at the end? This is unconventional, but also distracting as, like I said, I was often missing information, which were after included at the end.

Line-specific minor comments

L1: Since you are really focusing on seismic lines, why not just indicate that in your title?

L20: Always a source? Never a sink?

L25: Alberta probably was not as just a random example that you chose; presumably, you focused on it since these exploratory activities are so prominent there because of all the oil and gas present. I would make that clear.

L28: Increased from what to what and in what time frame? If you provide that info, it will give a better indication of the magnitude of the increase. Otherwise, it is hard to know if 4.4-5.1 kt should be considered a lot or not. (Or add "which is a ...% increase over the past ... years".)

L32: ... and planning of future exploratory activities.

L35: Globally?

L40: Maybe start this sentence with 'seismic' lines are'? To avoid starting with 'seismic exploration' for two sentences in a row.

L47: Move (LIS) before 'lines'.

L47: And elsewhere in Canada (BC, Yukon, NWT).

L51: I think the word 'ecosite' is mostly known and used in Canada. If this is meant to reach international audience, maybe use another word. Ecosystems? Or just 'wetlands'? (Also, you can add 'wetlands, which include peatlands').

L54: But if there are some studies, reference them here too, to be consistent with providing references for wildlife.

L54: Maybe give a brief definition of wetland vs. peatland.

I found these definitions later in the ms, but they would be useful here too.

L56: 'has been developed'... (?)

L57: '...under frozen ground conditions'....

'During' implies time, and 'conditions' is not time. If you want to use 'during' say 'during frozen ground season', or 'months', or something like that.

L61: persistent changes in vegetation

L63: I don't know to what depth you are referring to, but at depth 12.5 cm in peatlands I observed either no differences or that the lines were cooler than the sites adjacent to them. (But in the uplands they were significantly warmer).

L63-65: References?

(Williams and Quinton (Hydrol. Process. 27, 1854-65) observed that the latter is more important than the former.)

L76: Where is the Methods section?

L88: Total seismic line presence? Length?

L89: What do you mean by 'trails'? This is the first time you mention trails.

After reading the whole ms, I realized that you explain this better later on under Methods, but in that case, make a reference to it here.

L90: I don't know how you determined this but here is a sentence that I copied from an email from someone from ABMI: "the scale at which the ABMI captures seismic line information doesn't support the capture of LOW-IMPACT-SEISMIC category".

L101: So the big map shows the seismic lines and peatlands overlay? Maybe you should say that in the figure caption.

L107: I agree. It does miss LIS lines.

L101: This addresses some of my earlier concerns.

L114: So which one did you end up using? The publicly available or the more accurate one? If the former, then why?

L119: Out of curiosity, why would ABMI release underestimated version to the public and not the more accurate one?

L129: Are these estimated emissions truly 'resulting' from the WT position or is it just a correlation?

L130: Are the ranges of CH₄ fluxes 'resulting' from recent compilations? Maybe say 'based on'.

L137: I think that this should be stated in the previous paragraph, where you already talk about WT and emissions 'resulting' from it.

L138: Why transformed? Is it because it was not normally disturbed and/or the variance was not homogeneous?

L141: Where are these numbers coming from? I know you cite references, but a quick explanation (one sentence) of why you are using these particular numbers would be good here.

L152-153: I agree.

L157: Conduit? (Since 'plant community' is singular? Although community is a collective noun, so I'm not sure...).

L159: What do you mean by 'resulting conditions' here? Plant community? Environmental factors?

L166: What if one of these other factors, e.g. vegetation community type or changes in soil temperature, diminishes CH₄ emissions?

L184: What about Scandinavia?

L185: And other linear disturbances.

L188: Try not to overuse the word 'impact'. I was told by an editor that it is overused and not well-liked in scientific literature. You can use e.g. 'effects'.

L200: In the sentence above, you did not use a dash in 'land use'. Pick one and be consistent.

L224: Impact on what? Maybe 'peatland functioning' would be better here.

L241-243: Good that you recognize this.

L261: Above in this paragraph and earlier on in the ms, you already explained that LIS stands for Low Impact Seismic. So you don't have to spell it out again here.

L267: ... 'making it more sinuous'? (Otherwise you use the word 'line' or 'lines' six times in this sentence).

L267: Explain why is 'sinuous' important. E.g. because it better emulates natural patterns.

L268: Sounds like 'surveying' refers also to smaller drilling and vibrator equipment. Revise.

L271: Is this Table really necessary? You can provide that information in one sentence in the text. (Unless you want to have a quick reference to use elsewhere in the ms).

L291: Again, this is something I was aware of and concerned about, so good that you include that information.

L293: What assumptions?

L296: LIS lines.

Otherwise you are saying "in case of low impact seismic, some...", which sounds incomplete.

L310: If you are estimating emissions from Alberta peatlands, why use data from other places in western Canada?

L315: Same comment as above.

L324: What about emissions in winter months? Could this lead to further underestimations?

OK, you address this in the next paragraph.

L329: Which R package and which R version?

L342: This is an example of a case where the information, which I was looking for earlier on in the Results section, is included here at the end.

Anna Dabros

Reviewer #2 (Remarks to the Author):

Review of Strack et al. NatComm 2018, 'Petroleum exploration increases methane emissions in northern peatlands'

Strack et al. combine spatial data on peatland distribution and linear disturbance from (mostly seismic exploration lines) in Alberta to determine the area affected. They then combine this with regional mean methane fluxes, and flux response to elevated water table – a consequence of seismic line construction – to estimate the increase in methane emissions from this class of disturbance. This seems to be the first quantitative assessment of this disturbance impact on methane fluxes.

These linear disturbance segments are 1-10 m wide and roughly 1 to tens of km long (based on their Table 2 and Fig. 3) so they are difficult to map over large regions (e.g., Alberta) with satellite imagery.

The paper is clearly written, and the figures and tables are generally clear and useful.

I HAVE 2 MAJOR QUESTIONS:

SIGNIFICANCE: To make comparisons easier, please report all methane fluxes in same units, at least somewhere in the document; you have 't' (table 1), 'kt' (text for seismic lines) and 'Tg' (for

Alberta total peatlands, and global northern peatlands. If I converted correctly, 4-5 kt methane from seismic lines amounts to about 1-2% of Alberta total peatland methane flux, and 0.015% of northern peatland total methane flux. This is not a huge flux impact, though it is also not zero; as you acknowledge, there are reasons why you estimate may be low, but you don't really provide any quantification of those factors. I agree that this should be reported in Canada GHG inventory, in principle, but the uncertainty in your estimation is still quite high and the impact isn't large, so I could understand reluctance to do so.

Where is Conklin on the Fig. 2 map (which hue and intensity)? Is the 10000 km² comparison area about 10x10 of the hexagons in fig. 2? Is there any way to know if the human footprint inventory in the Conklin region is generalizable (e.g., what fraction is LIS in that region and is it typical? is there a way to show on Fig. 3 which lines are LIS? Are any of the linear features missed in the human footprint inventory lines that were constructed after the inventory? How dynamic is this – what is a rough rate of seismic line construction (km/y) and how has that varied? In the end, I think this is an interesting study, and addresses an under-studied disturbance that is generally not accounted for. It is also important that changes in practice – i.e., to LIS – has probably reduced the impact, but I don't come away from the article with a good sense for how strong that impact has been (see next point).

MODEL/ALGORITHM FOR ESTIMATION: Are you confident that the only difference from a methane point of view between legacy and LIS lines (per km of disturbance) is width of impact (i.e., a factor of 2 in Table 2)? Is vegetation impact/change different? Temperature regime change? Degree of compaction? You mention these factors as important for methane fluxes, but don't address them in the model. E.g., discussion paragraph 1: Strack et al. (2018) attributed order of magnitude increase in methane to higher soil temperature and shift in plant community. Yet the model in this study uses a change in WT to estimate methane impact. Why the disparity? Shifts in hydrology are likely related to or correlated with shifts in temperature and vegetation communities, but does it capture all effects. You note that 'seismic lines in peatlands persist for decades'¹¹ [line 202]. Does the impact on methane emissions remain constant over that time (as your model assumes) or does it change as the lines 'recover'? How many decades, 1 to 2, or several to many?

SOME ADDITIONAL QUESTIONS OR CLARIFICATIONS

Lines 46-50: Are low-impact seismic (LIS) lines also cleared by bulldozer to bare soil?

Fig. 1: does the impact of these lines spread laterally to any significant degree, effectively widening their swath of impact? Do they affect water flow that is perpendicular to the lines, leading to wetter 'upstream' and drier 'downstream' sides? Would the soil warming impact be different for N-S vs. E-W lines, due to shading ('more direct insolation') from low sun angles (in Fig. 3, most lines seem to be oriented NS or EW) Is there discontinuous or sporadic or isolated permafrost in northern Alberta; do these seismic lines lead to enhance thaw/thermokarst?

Table 1. You should specify in the table how CO₂-equivalent emissions were computed. (Is this conversion really necessary?)

Table 2 isn't really needed, it could just be a sentence.

Line 87: Would freshwater swamps without peat (<40 cm) still be significant methane sources?

Fig. 4: I suggest adding vertical lines for the mean values for fen bog and swamp, and for increase due to elevated WT.

Line 179: 0.46 km per km – should that be per km-squared? Awkward place to have a numerical citation.

Line 263: older seismic line width 4-8 m; line 46: older seismic line width 6-10 m; which is it?

Line 290: 'many narrow lines ...' Can you be a bit more specific or clear about what you mean by many?

Table S3. What are footnotes 3 and 4 on 'Seismic line length from ABMI Human Footprint database³' and 'Seismic line length from ABMI Enhanced Linear Feature database⁴'? Are those referring to citations 1 and 2?

-Steve Froking

Reviewer #1 (Remarks to the Author):

General comments for Authors

The study addresses the effects of oil and gas exploratory activities (through construction of seismic lines) on methane emissions in northern peatlands. The place of study is in the province of Alberta, Canada, where both peatlands and seismic lines are highly abundant. As such, Alberta is a good choice for this study, but I also think that the fact that Alberta is so rich in oil and gas reserves (third highest in the world) and so densely disturbed by seismic lines, also makes it not necessarily an accurate representative of what is happening in the rest of Canada (or the rest of northern hemisphere for that matter). This should be recognized, if the study is to be used in providing information to be used at Canadian and/or global scale (e.g. for reporting GHG emissions). Still, I think that the study addresses a very important matter, which has not been adequately addressed before. I agree that there is not a lot of research (or the results of it are not easy to find/access), which addresses the effects of seismic lines on C, and GHG emissions in particular. Considering the abundance of northern peatlands, their importance as C sinks/sources, and the intensity of industrial disturbances on these peatlands, (including seismic lines, which are among the most prominent industrial linear disturbances), I think that the subject of this study is highly pertinent in the accurate assessment of GHG emissions and their effect on global climate change.

We thank the reviewer for recognizing the importance of this topic, and for her careful and thoughtful review.

We agree that it is important readers are not left with the impression that the disturbance regime in Alberta is representative of the situation across all northern peatlands; this was never our intent. We devote a whole paragraph to this topic in the Discussion (para 2), but don't do as good a job of this in the abstract. As a result, we have made a couple of alterations:

- In the abstract, the line "In the northern hemisphere, extensive geologic-exploration activities have occurred to locate and map oil and gas reservoirs" has been re-written: "In the northern hemisphere, extensive geologic-exploration activities have occurred in locations where petroleum deposits are present".
- Again in the abstract, the line "Using the Canadian province of Alberta as an example, we mapped the occurrence of land disturbed by oil and gas exploration on peatlands and estimated the induced CH₄ emissions" has been re-worded: "We mapped petroleum-exploration disturbances on peatlands in the Canadian province of Alberta, where peatlands and oil deposits are widespread, and estimated the induced CH₄ emissions"

The methodology of the study has some shortcomings, which were nevertheless recognized and discussed (e.g. underestimation of the amount of LIS lines). Yet, generally I think that the resources available were used to the best of their potential. However, the study is based on previously known relationship of water table to CH₄ emissions only. While this relationship may be accurate enough, other important factors that have not been considered (soil temperature, plant community etc.) could have a different (even opposite) effects, which may potentially mitigate the effects of increased water table under seismic lines. Actual field measurements on and off seismic lines (including consideration of other than only water table factors) seem like the next logical step to corroborate the results of this study.

This is a fair comment, and as noted by the reviewer, we do highlight that these factors have not been included. However, in most cases, we hypothesize that changes in other factors (such as temperature and plant community) will result in even greater CH₄ emissions than we predict, such that our estimates are conservative.

We appreciate that more emphasis could be placed on highlighting the uncertainty in our modelled relationship, and have addressed this with the following alterations:

In the abstract we have added/reworded several sentences and it now reads “However, uncertainty in these emissions remains large due to the dearth of measurements of environmental conditions and GHG fluxes from peatlands affected by petroleum exploration, limiting our process-based understanding of ecosystem response. Given the extent of peatland areas affected, research programs further investigating the potential effects of petroleum exploration on peatland GHG fluxes are required to allow for appropriate consideration of these emissions in the planning of future exploratory activities and peatland restoration programs.”

In the discussion we have better justified our choice to focus on the WT relationship saying “We have chosen to use WT position to estimate the potential impact as the shift in WT on peatland seismic lines has been reported for both bogs²¹ and fens²⁰. The former study mapped average WT changes over a 61 ha area, providing more confidence that these represent more than local conditions.”

Statistical methods seem appropriate, though more precise information on which R version, package(s) and functions were used would be appreciated. Explanation of why the CH₄ flux and WT data was log transformed (presumably because of normality and homogeneity of variance assumptions violations?) would be good too. Supplementary information provided valuable additional information.

This has been added to the methods. The log transformation greatly improved normality and homogeneity of residuals.

Overall, the manuscript is well-written: concise but comprehensive. However, as I was reading

the manuscript, in quite a few instances I had a question/concern in the Results section, which was then answered/addressed later on, under the Methods section. Why is the Methods section at the end? This is unconventional, but also distracting as, like I said, I was often missing information, which were after included at the end.

We appreciate this concern, but the manuscript is organized in the order required by the journal. We have added references to the methods in the results, when appropriate, to help the reader recognize that more information is coming.

Line-specific minor comments

L1: Since you are really focusing on seismic lines, why not just indicate that in your title?

This is a fair point, but we are not convinced that readers will know what we mean by seismic lines prior to reading the paper. We have decided to leave the title unchanged.

L20: Always a source? Never a sink?

Locations within a peatland can act as a CH₄ sink, but on a whole, they act as a source. As this is a general statement, it is correct and no changes have been made.

L25: Alberta probably was not as just a random example that you chose; presumably, you focused on it since these exploratory activities are so prominent there because of all the oil and gas present. I would make that clear.

We have altered the text in the Abstract to make it clear why this region was used. The passage now reads “We mapped petroleum-exploration disturbances on peatlands in the Canadian province of Alberta, where peatlands and oil deposits are widespread...”.

L28: Increased from what to what and in what time frame? If you provide that info, it will give a better indication of the magnitude of the increase. Otherwise, it is hard to know if 4.4-5.1 kt should be considered a lot or not. (Or add "which is a ...% increase over the past ... years".)

We have updated the abstract to clarify that these estimated emissions are increases “from undisturbed conditions”. We have also added in the abstract that including this emissions would increase CH₄ emissions related to land use, land use change and forestry by 7-8%.

L32: ... and planning of future exploratory activities.

We have added the suggested text.

L35: Globally?

Yes. We have updated the text to clarify. The passage now reads “Globally, northern peatlands cover ~4 million km² and store 500 ± 100 Gt of soil carbon”.

L40: Maybe start this sentence with 'seismic' lines are'? To avoid starting with 'seismic exploration' for two sentences in a row.

We haven't introduced the term “seismic lines” at this point, so instead, we altered the structure of the following sentence so that it doesn't also start with seismic exploration. The second passage now reads “In Alberta, seismic exploration has been ongoing since 1929...”

L47: Move (LIS) before 'lines'.

Done.

L47: And elsewhere in Canada (BC, Yukon, NWT).

We've added this information. The passage now reads “Since then, these ‘legacy’ lines have been replaced by ‘low-impact’ seismic (LIS) lines in Alberta and elsewhere in Canada (e.g., British Columbia, Yukon, Northwest Territories), with a reduced width...”

L51: I think the word 'ecosite' is mostly known and used in Canada. If this is meant to reach international audience, maybe use another word. Ecosystems? Or just 'wetlands'? (Also, you can add 'wetlands, which include peatlands').

Good point. We've changed this to “wetland ecosystems”. In the next paragraph we clearly indicate that most wetlands in Alberta are peatlands, so we haven't added the “which includes peatlands” here to avoid redundancy.

L54: But if there are some studies, reference them here too, to be consistent with providing references for wildlife.

Done

L54: Maybe give a brief definition of wetland vs. peatland. I found these definitions later in the ms, but they would be useful here too.

We have added the Canadian peatland definition here for clarity. It is given very generally here and therefore does not overlap the information given later in the manuscript.

L56: 'has been developed'... (?)

We have added “been” here.

L57: '...under frozen ground conditions'....

'During' implies time, and 'conditions' is not time. If you want to use 'during' say 'during frozen ground season', or 'months', or something like that.

Changed to “on frozen ground”.

L61: persistent changes in vegetation

Changed.

L63: I don't know to what depth you are referring to, but at depth 12.5 cm in peatlands I observed either no differences or that the lines were cooler than the sites adjacent to them. (But in the uplands they were significantly warmer).

The references given refer to shallow soils (5 cm) in a non-permafrost peatland and thawing permafrost. In the latter, this would result in warmer soils through the depths that were previous frozen (so could extend to 1 m or more). We think this statement fairly summarizes literature that is available, although we note that it is limited.

L63-65: References?

(Williams and Quinton (Hydrol. Process. 27, 1854-65) observed that the latter is more important than the former.)

References have been added.

L76: Where is the Methods section?

It follows the discussion as this is the required structure for this journal

L88: Total seismic line presence? Length?

We were referring to length. We added a parenthesis to the sentence, with reference to Methods, where we describe this data more fully. The passage now reads: “Based on these data sets, we estimate that at least 345,000 km of seismic lines and trails (measured here as length; see Methods for complete definitions) cross peatlands in Alberta, of which about 10% are LIS (Table 1)”.

L89: What do you mean by 'trails'? This is the first time you mention trails.

After reading the whole ms, I realized that you explain this better later on under Methods, but in that case, make a reference to it here.

Good point. These are defined in the methods section, so we have added a reference here to the methods section for clarity. This is part of the parenthesis quoted in the previous comment.

L90: I don't know how you determined this but here is a sentence that I copied from an email from someone from ABMI: "the scale at which the ABMI captures seismic line information doesn't support the capture of LOW-IMPACT-SEISMIC category".

We agree and this is explored later in the manuscript. Here we refer to the lines defined as low impact seismic in the ABMI dataset that we used. We believe that to a reader not familiar with the details of the datasets, this statement will not be out of place and the issue of actually mapping LIS will be clear once they read the whole manuscript. We have not made any changes here.

L101: So the big map shows the seismic lines and peatlands overlay? Maybe you should say that in the figure caption.

We have supplemented the figure caption accordingly. It now reads "Peatland and seismic line density across the province of Alberta. Hot colors indicate locations where high densities of seismic lines and peatlands coincide."

L107: I agree. It does miss LIS lines.

No changes

L101: This addresses some of my earlier concerns.

Good.

L114: So which one did you end up using? The publicly available or the more accurate one? If the former, then why?

While the ABMI Enhanced Linear Feature layer is more accurate, it is not available for the entire province. We performed our analysis using the best-available (and complete) data set.

L119: Out of curiosity, why would ABMI release underestimated version to the public and not the more accurate one?

The ABMI performs its human-footprint mapping using satellite imagery with ~10m pixels. As a result, LIS do not show up reliably. The Enhanced Linear Features layer was completed using high-resolution air photos over a limited portion of the Lower Athabasca. While the publically released ABMI human footprint inventory has its limitations, it is still the best source of province-wide information.

L129: Are these estimated emissions truly 'resulting' from the WT position or is it just a correlation?

We mean that based on the estimated mean WT, we calculated this CH₄ emission. We've changed the wording to "leading to".

L130: Are the ranges of CH₄ fluxes 'resulting' from recent compilations? Maybe say 'based on'.

We've changed the text to "reported in"

L137: I think that this should be stated in the previous paragraph, where you already talk about WT and emissions 'resulting' from it.

Good point. We moved the result about the statistical significance to the previous paragraph.

L138: Why transformed? Is it because it was not normally distributed and/or the variance was not homogeneous?

As in response to an earlier comment, we've added more details about this to the methods section.

L141: Where are these numbers coming from? I know you cite references, but a quick explanation (one sentence) of why you are using these particular numbers would be good here.

These are values previously reported in literature and we've clarified this in the text now. This is also clarified in the Methods section, so we have also added reference to the Methods in the text here too. The passage now reads "Starting with the mean WT estimated for each peatland type (Figure 4), we then moved the mean WT position closer to the surface on the footprint of the seismic lines according to the change in WT previously reported in literature (13.9 to 15.4 cm^{19,20}) and re-calculated CH₄ flux (see Methods for more details)".

L152-153: I agree.

No changes required

L157: Conduit? (Since 'plant community' is singular? Although community is a collective noun, so I'm not sure...).

We have considered singular (as the community) in this sentence, so we've changed this to "a conduit"

L159: What do you mean by 'resulting conditions' here? Plant community? Environmental factors?

We have edited the text to improve clarity. The passage now reads "Other studies have shown width, age, and orientation to influence thermal and hydrological conditions on linear disturbances"

L166: What if one of these other factors, e.g. vegetation community type or changes in soil temperature, diminishes CH4 emissions?

It's possible, but in most cases, literature suggests that changes would all result in higher CH4 flux. No additional changes made.

L184: What about Scandinavia?

We have not found any literature on seismic exploration in Scandinavia and asked several colleagues who also report that this is not an important disturbance in that region. We have not made any changes to the text.

L185: And other linear disturbances.

Good point. We've added "along with other linear disturbances such as winter roads" to the text here.

L188: Try not to overuse the word 'impact'. I was told by an editor that it is overused and not well-liked in scientific literature. You can use e.g. 'effects'.

Thanks for the suggestion. We've changed this instance to effect.

L200: In the sentence above, you did not use a dash in 'land use'. Pick one and be consistent.

We had meant to use "land use" throughout. We've corrected this occurrence and checked the rest of the manuscript for consistency.

L224: Impact on what? Maybe 'peatland functioning' would be better here.

We agree that this was unclear. We altered the text here to read "highlighting the need for improved understanding of the effects of seismic lines on peatland function prior to the extensive application of restoration techniques"

L241-243: Good that you recognize this.

No changes required

L261: Above in this paragraph and earlier on in the ms, you already explained that LIS stands for Low Impact Seismic. So you don't have to spell it out again here.

Changed

L267: ... 'making it more sinuous'? (Otherwise you use the word 'line' or 'lines' six times in this sentence).

We've edited this sentence to improve readability and reduce the use of the word "line". The passage now reads "Seismic programs using the LIS construction methods have significantly narrower disturbances and often use avoidance construction methods and/or hand-cut lines to reduce not only the width of the line but also making them more sinuous to reduce line of site and better emulate natural patterns".

L267: Explain why is 'sinuous' important. E.g. because it better emulates natural patterns.

We've added this to the sentence. Sinuosity also reduces the line of site for predators. Passage is quoted above.

L268: Sounds like 'surveying' refers also to smaller drilling and vibrator equipment. Revise.

We have revised the sentence. The passage now reads "This is possible in part due to advances in surveying with GPS, and also from the use of smaller drilling and vibrator equipment".

L271: Is this Table really necessary? You can provide that information in one sentence in the text. (Unless you want to have a quick reference to use elsewhere in the ms).

We have removed the table and now list the average widths in the last sentence of this paragraph.

L291: Again, this is something I was aware of and concerned about, so good that you include that information.

No changes required

L293: What assumptions?

The original text was awkwardly phrased – *assumptions* is not the correct term. We have revised the passage to read as follows: "The distinction between trails, LIS and legacy seismic line features is based on human interpretation. Therefore, misclassification may have occurred in some cases."

L296: LIS lines. Otherwise you are saying "in case of low impact seismic, some...", which sounds incomplete.

Fixed

L310: If you are estimating emissions from Alberta peatlands, why use data from other places in western Canada?

We were trying to build a database that represented the climate conditions and peatland types of Alberta, but brought together additional data sources. If only Alberta was used, there would be data from <10 peatlands, most of which was collected by the authors of this paper. We have added a sentence to explain this.

L315: Same comment as above.

Not sure quite what the reviewer means here, but for the literature search, we used “Canada” as a search term, but only included data within our geographic bounds as defined in the previous sentences. For swamps, we did include all Canadian sites as there were very few studies in western Canada and this has all been described in the methods.

L324: What about emissions in winter months? Could this lead to further underestimations? OK, you address this in the next paragraph.

No further changes made.

L329: Which R package and which R version?

We didn't use any additional R packages not included in the main R release. The reference to the R version is given here (i.e., the number listed links to the R reference in the literature cited section).

L342: This is an example of a case where the information, which I was looking for earlier on in the Results section, is included here at the end.

As noted above, this is due to the required format of the paper. In response to the earlier comment, we have added a reference to the methods section in the results where we use this information so that the reader will now know to refer to the methods for more details.

Anna Dabros

Reviewer #2 (Remarks to the Author):

Review of Strack et al. NatComm 2018, 'Petroleum exploration increases methane emissions in northern peatlands'

Strack et al. combine spatial data on peatland distribution and linear disturbance from (mostly seismic exploration lines) in Alberta to determine the area affected. They then combine this with regional mean methane fluxes, and flux response to elevated water table – a consequence of seismic line construction – to estimate the increase in methane emissions from this class of disturbance. This seems to be the first quantitative assessment of this disturbance impact on methane fluxes.

These linear disturbance segments are 1-10 m wide and roughly 1 to tens of km long (based on their Table 2 and Fig. 3) so they are difficult to map over large regions (e.g., Alberta) with satellite imagery.

The paper is clearly written, and the figures and tables are generally clear and useful.

I HAVE 2 MAJOR QUESTIONS:

SIGNIFICANCE: To make comparisons easier, please report all methane fluxes in same units, at least somewhere in the document; you have 't' (table 1), 'kt' (text for seismic lines) and 'Tg' (for Alberta total peatlands, and global northern peatlands. If I converted correctly, 4-5 kt methane from seismic lines amounts to about 1-2% of Alberta total peatland methane flux, and 0.015% of northern peatland total methane flux. This is not a huge flux impact, though it is also not zero; as you acknowledge, there are reasons why you estimate may be low, but you don't really provide any quantification of those factors. I agree that this should be reported in Canada GHG inventory, in principle, but the uncertainty in your estimation is still quite high and the impact isn't large, so I could understand reluctance to do so.

Our conservative estimate of methane from seismic lines in Alberta (4.4-5.1 kt) is 7-8% of that reported for land use, land use change and forestry across all of Canada (63 kt). We suggest that this is the more appropriate comparison.

We agree with the reviewer that there is a lot of uncertainty in our estimate, and that this element should be highlighted more in the text. The first version of the manuscript discussed our results primarily as they relate to industry and regulators; the current version includes alterations designed to expand our call to the research community to work on reducing these uncertainties:

To highlight these issues more clearly (and also in response to Reviewer 1) we have added the following to the abstract:

"However, uncertainty in these emissions remains large due to the dearth of measurements of environmental conditions and GHG fluxes from peatlands affected by petroleum exploration, limiting our process-based understanding of ecosystem response."

In the discussion we now also note:

"While our estimate of peatland seismic line impact on CH₄ emissions, 4.4 – 5.1 kT yr⁻¹ is small compared to the ~4000 kT yr⁻¹ of CH₄ emitted from all Canadian peatlands⁴¹, the present study represents emissions only from the province of Alberta and this total is not negligible in terms of national CH₄ emissions arising from land use"

Where is Conklin on the Fig. 2 map (which hue and intensity?)?

We have added Conklin to the Figure 2 map - good suggestion. It is red.

Is the 10000 km² comparison area about 10x10 of the hexagons in fig. 2?

The hexagons in Figure 2 are 100 km², so yes. We have edited the caption of Figure 3 to include this information.

“For the sake of comparison, the hexagons in Figure 1 (which includes the town of Conklin) are 100 km² each.”

Is there any way to know if the human footprint inventory in the Conklin region is generalizable (e.g., what fraction is LIS in that region and is it typical?)

We could only speculate on this, and so will refrain from doing so in the manuscript.

is there a way to show on Fig. 3 which lines are LIS?

We experimented with this, but the map is so dense that additional symbology makes it confusing. Since the primary purpose of the map is to show the difference between the public dataset and enhanced dataset, we have elected to leave the figure as-is. However, we have added a sentence to the caption addressing this, since distinguishing LIS lined from conventional is pretty straightforward to the trained eye.

“While LIS lines are not specifically symbolized on the map, they are relatively easy to distinguish from conventional networks on account of their very high density: often spaced less than 100 m apart.”

Are any of the linear features missed in the human footprint inventory lines that were constructed after the inventory?

Once again, we could only speculate on this.

How dynamic is this – what is a rough rate of seismic line construction (km/y) and how has that varied?

This is a very good question, but ultimately outside the scope of this manuscript. We currently lack the data to address this at the provincial scale. Most contemporary seismic disturbances are LIS, which – as we document – are under-represented in publically available data.

In the end, I think this is an interesting study, and addresses an under-studied disturbance that is generally not accounted for. It is also important that changes in practice – i.e., to LIS – has probably reduced the impact, but I don't come away from the article with a good sense for how strong that impact has been (see next point).

MODEL/ALGORITHM FOR ESTIMATION: Are you confident that the only difference from a methane point of view between legacy and LIS lines (per km of disturbance) is width of impact (i.e., a factor of 2 in Table 2)? Is vegetation impact/change different? Temperature regime

change? Degree of compaction? You mention these factors as important for methane fluxes, but don't address them in the model. E.g., discussion paragraph 1: Strack et al. (2018) attributed order of magnitude increase in methane to higher soil temperature and shift in plant community. Yet the model in this study uses a change in WT to estimate methane impact. Why the disparity? Shifts in hydrology are likely related to or correlated with shifts in temperature and vegetation communities, but does it capture all effects.

This is an excellent point and we hope to be able to apply a more process-based model of changes in the future. At this point, field data is very limited, not only on the resulting fluxes, but also on the shifts in environmental conditions. In previous studies, we have been able to map the average WT change on seismic lines across a 61 ha area. Although this is still a very small area, we have more confidence in using this to estimate change in WT (which is also generally well correlated to plant community) generally on seismic lines. Moreover, we are more confident that geospatial methods will be able to map shifts in WT and plant communities in the future to improve estimates. While we also think temperature is important, we have found data in literature only from one non-permafrost peatland site (i.e., Strack et al., 2018), and have no easy way to map this across large areas. The fact that temperature changes will also be linked to WT changes and management on the line (e.g., whether debris piles are left on site; Williams and Quinton, 2013) further complicate applying an average temperature increase across the province.

We've updated the discussion, to justify our choice of using WT, but also to call more clearly for the need for future research. The first few sentences of the discussion now read:

"We have chosen to use WT position to estimate the potential impact as the shift in WT on peatland seismic lines has been reported for both bogs²¹ and fens²⁰. The former study mapped average WT changes over a 61 ha area, providing more confidence that these represent more than local conditions. However, seismic lines likely also alter thermal and ecological conditions (Figure 1), which are known controls on peatland CH₄ emissions."

And later in the paragraph add:

"It is clear that more research is needed to better quantify actual changes in CH₄ emissions under the variety of disturbance conditions that occur on peatland seismic lines and enable future work to estimate emissions using process-based models."

You note that 'seismic lines in peatlands persist for decades'¹¹ [line 202]. Does the impact on methane emissions remain constant over that time (as your model assumes) or does it change as the lines 'recover'? How many decades, 1 to 2, or several to many?

This is an excellent question but at this point there is actually no field data available to answer this. We are hoping that results from this paper will drive some of that data collection and we are working on it ourselves, but at this time, the persistence of the shifts in CH₄ flux remains unknown. We mention this in the discussion.

SOME ADDITIONAL QUESTIONS OR CLARIFICATIONS

Lines 46-50: Are low-impact seismic (LIS) lines also cleared by bulldozer to bare soil?

Based on this comment, we realized that we needed to provide more information on seismic line construction/disturbance in this section. We have added the following sentence that should clarify impacts to soils on seismic lines, including reference to a review paper:

“Although construction of lines prior to 1960 often resulted in substantial soil disturbance, improved management practices, including LIS, has greatly reduced the damage to soil and ground layer vegetation¹⁰”

Fig. 1: does the impact of these lines spread laterally to any significant degree, effectively widening their swath of impact? Do they affect water flow that is perpendicular to the lines, leading to wetter ‘upstream’ and drier ‘downstream’ sides? Would the soil warming impact be different for N-S vs. E-W lines, due to shading (‘more direct insolation’) from low sun angles (in Fig. 3, most lines seem to be oriented NS or EW) Is there discontinuous or sporadic or isolated permafrost in northern Alberta; do these seismic lines lead to enhance thaw/thermokarst?

So far we don’t have any clear evidence of the impact spreading very far into the adjacent peatland or that compression blocks the flow of water (Strack et al., 2018), but there are edge effects reported for seismic lines in uplands (Dabros et al., 2017) and we have mentioned this in the discussion. This is definitely an area where more work is needed and that we’re working on moving it forward. The direction of the line is also important for insolation and in permafrost area, seismic lines have been clearly shown to act as initiation point for permafrost thaw. These are all areas requiring further research and we include them all in the discussion. There are some areas of sporadic permafrost in the very northern parts of AB, but the majority of lines will be on non-permafrost peatlands. For this reason, we have chosen not to include this aspect explicitly in Fig. 1.

Table 1. You should specify in the table how CO2-equivalent emissions were computed. (Is this conversion really necessary?)

A good point and we agree that the conversion isn’t really needed. Therefore, we’ve removed it from the table and the manuscript text. This also avoids debate over which global warming potential value should be chosen for the conversion.

Table 2 isn’t really needed, it could just be a sentence.

This has been removed and is now listed at the end of the preceding paragraph. Reviewer 1 made a similar comment and we agree that the table is not necessary

Line 87: Would freshwater swamps without peat (<40 cm) still be significant methane sources?

Yes, it is likely and some of the data we've compiled from literature has been derived from mineral soil swamps. To better describe the impact this has on our estimates, we added a section to the discussion that reads:

"We also included swamps in our estimates of seismic line impacts to peatlands, although some are likely mineral soil wetlands. Assuming both mineral soil and peatland swamps responds similarly to linear disturbances, our estimate of increased CH₄ emissions from swamps would not overestimate the impact in emissions, but may misclassify some wetland impact as specific to peatlands. As there is no available data on the hydrologic impact of seismic lines in swamps, the potential uncertainty of including mineral soil swamps in our estimates is unclear. However, studies of tree regrowth on seismic lines also indicate poor recovery in swamp ecosystems¹¹, suggesting that including them in our analysis is warranted."

Fig. 4: I suggest adding vertical lines for the mean values for fen bog and swamp, and for increase due to elevated WT.

We have added vertical lines for the mean values calculated from the data for each peatland class (i.e., mean WT in undisturbed condition). As we calculate emissions based on 2 different increases in the WT position based on available studies, this would require adding an additional 6 lines to the figure (so 9 vertical lines total). Visually, this creates a very messy plot. Therefore, we have only chosen to show the mean and provide the reader with the increase we used in the caption.

Line 179: 0.46 km per km – should that be per km-squared? Awkward place to have a numerical citation.

It should be per square km and we agree that the numerical reference is in an awkward position here (and likely led to the deletion of the 2 in m²). We have reorganized this sentence and corrected the units.

Line 263: older seismic line width 4-8 m; line 46: older seismic line width 6-10 m; which is it?

This varies in the literature and the 4-8 m reference here is specific to the database we used. We acknowledge that having 2 different values is misleading, so we've now used the range listed in Dabros et al., 2018 (5-10 m) in both places in the text for consistency and then listed the mean width for legacy lines (6 m) that we actually used in the area calculation.

Line 290: 'many narrow lines ...' Can you be a bit more specific or clear about what you mean by many?

We're referring to LIS lines, and have altered the text to be more specific. The passage now reads "SPOT6 images are unable to locate narrow features (<~1.5 m). Therefore, many LIS lines that are present on the ground are not included in this dataset".

Table S3. What are footnotes 3 and 4 on 'Seismic line length from ABMI Human Footprint

database3' and 'Seismic line length from ABMI Enhanced Linear Feature database4'? Are those referring to citations 1 and 2?

Yes, this was our error. These should be 1 and 2 as numerical citations to the references given in the supplemental material.

-Steve Frolking

REVIEWERS' COMMENTS:

Reviewer #1 (Remarks to the Author):

The authors have addressed all the comments I made for the first submitted draft of the manuscript, and I am satisfied with their justifications and the changes made. Well done.

A few minor additional comments:

L 29: In the Abstract, when you talk about 7-8% increase in emissions, is it an increase in comparison to today's levels, or some other time? Please clarify.

L 79: When you talk about your calculations resulting in 1900 km² of peatlands disturbed by seismic lines in AB, add a reference to where you explain how you made these calculations.

L 284-287 and 299: When you refer to Table 2, sometimes you say Supplementary Table 2 and sometimes just Table 2. Are they the same Table? I'm assuming that yes, since I could only find Supplementary Table 2 in your files, but just to be clear, make it consistent.

Anna Dabros

Reviewer #2 (Remarks to the Author):

The authors' response to the two reviews was very thorough and clear. They have addressed all of my review comments, as well as those of the other reviewer. I consider the paper now suitable for publication in Nature Communications.

-Steve Frolking

Reviewers provided a few minor comments and our responses are below:

L 29: In the Abstract, when you talk about 7-8% increase in emissions, is it an increase in comparison to today's levels, or some other time? Please clarify.

Response: We have added the word "current" to clarify we meant an increase over today's emissions.

L 79: When you talk about your calculations resulting in 1900 km² of peatlands disturbed by seismic lines in AB, add a reference to where you explain how you made these calculations.

Response: We have added "(see details in methods)"

L 284-287 and 299: When you refer to Table 2, sometimes you say Supplementary Table 2 and sometimes just Table 2. Are they the same Table? I'm assuming that yes, since I could only find Supplementary Table 2 in your files, but just to be clear, make it consistent.

Response: Yes, this is the same table and this was missed in the revision as this table was moved to supplementary information. We have checked all table references carefully and corrected this error.